# The Establishment of Evaluation Models for the Cooking Suitability of Different Pork Muscles

**DOI:** 10.3390/foods12040742

**Published:** 2023-02-08

**Authors:** Shengnan Duan, Xiaoyan Tang, Junliang Zhan, Suke Liu, Yuhui Zhang

**Affiliations:** 1Key Laboratory of Meat Processing and Quality Control, College of Food Science and Technology, Nanjing Agricultural University, Nanjing 210095, China; 2Key Laboratory of Agrifood Safety and Quality, Ministry of Agriculture, Institute of Quality Standards and Testing Technology for Agro-Products, Chinese Academy of Agricultural Sciences, Beijing 100081, China

**Keywords:** pig meat, different parts, suitable cooking method, key quality indicators, predictive models

## Abstract

Pork is the main meat consumed by Chinese people. In this study, the sensory quality of four muscles (loin, shoulder, belly, and ham) under three cooking methods (boiling, scalding, and roasting) was examined, and the edible quality and nutritional quality of fresh meat were determined at the same time. Principal component analysis, cluster analysis, correlation analysis, and analysis of the coefficient of variation were used to determine key quality indicators, from which comprehensive quality evaluation equations were established. The results showed that, when meat was boiled, the comprehensive quality evaluation model was $Y=0.1537X_{1}+0.1805X_{2}+0.2145X_{3}+0.2233X_{4}+0.2281X_{5}$ (*X*_1_~*X*_5_ are *a**, fat, odor, tenderness, and flavor, respectively) and the most suitable muscle was belly; when meat slices were scalded in a hot pot, the comprehensive quality evaluation model was $Y=0.1541X_{1}+0.1787X_{2}+0.2160X_{3}+0.2174X_{4}+0.2337X_{5}$ (*X*_1_~*X*_5_ are *a**, fat, odor, tenderness, and flavor, respectively) and the most suitable muscle was belly; when meat was roasted, the comprehensive quality evaluation model was $Y=0.1539X_{1}+0.1557X_{2}+0.1572X_{3}+0.1677X_{4}+0.1808X_{5}+0.1845X_{6}$ (*X*_1_~*X*_6_ are flavor, marbling, elasticity, cooked flesh color, tenderness, and flesh color, respectively), and the most suitable muscles were belly and shoulder.

## 1. Introduction

Pork has an important place in the human diet [1]. According to the latest forecast of Economic Cooperation and Development (OECD) and the Food and Agriculture Organization of the UN(FAO) for 2031 [1,2], the global pork consumption demand averaged 110.5 million metric tons from 2019 to 2021. China is a major pork market, accounting for more than 40% of the world’s pork consumption according to data released by the OECD and FAO. In 2021, global pork production was 106.1 million tons, while China accounted for more than 43% of the total global pork production [3].

Pork has a long history in China. Since the early development of human society, pork has entered the human diet, which reflects the important position of pork in people’s food culture. Pork is the main source of meat for the Chinese people because of its low price, rich oil and high nutrition, which can replenish energy for the human body. There are many traditional pork cooking methods in China, such as steaming, boiling, roasting, frying, and scalding in a hot pot, which can produce good meat flavor and are conducive to human digestion and absorption [4]. For example, roasted pork produces a unique barbecue flavor [5].

Different muscles of pork have varied edible quality, nutritional quality, and processing quality, and the most suitable cooking methods for them are also varied. When different cooking methods are adopted for these muscles, they produce different flavors [6]. To improve life quality, scholars begin to pay more attention to the suitability of meat processing and cooking methods for better flavor and enjoyment. At present, meat processing suitability research mainly focuses on beef and mutton [7,8,9]. For example, it is found that high rib, silverside, striploin, tenderloin, and topside of beef are more suitable for roasting [10], and sirloin, knuckle, and beef tendon are suitable for stewing [11]; shank striploin, knuckle, loin, and silverside of mutton are suitable for roasting, loin, shank, striploin, knuckle, chuck tender, and silverside are suitable for boiling, and loin, striploin, and shank are suitable for scalding [12]. However, there is no relevant report on the cooking and processing suitability of pork muscles, revealing that research on the comprehensive evaluation model for the cooking suitability of different muscles of pork is relatively lacking. Therefore, building a comprehensive evaluation model for the cooking suitability of pork muscles can provide scientific guidance to consumers.

In the process of establishing an evaluation model, there are four common methods for assigning indicator weights: analytic hierarchy process, principal component analysis (PCA), expert scoring, and fuzzy comprehensive evaluation. PCA avoids the subjective influence of experts and helps transform multiple complex indicators into a few comprehensive indicators by dimension reduction [13].

In this study, loin, ham, shoulder, and belly from Duroc × (Landrace × Yorkshire) pigs (DLYs) were used to analyze the differences in the edible quality and nutritional quality of fresh meat, and the sensory scores under three cooking methods including boiling, scalding, and roasting. Key quality indexes of pork under different cooking methods were determined, and an evaluation model was established to evaluate the cooking suitability of different muscles of pork, so as to provide guidance on the cooking of different muscles of pork.

## 2. Materials and Methods

### 2.1. Materials

Pork muscle samples from six 180-day-old DLY pigs (females) of 80 kg carcass weight fed by the same feeding method were collected from Beijing Shunxin Pengcheng Co., Ltd. (Beijing, China). The slaughtered pigs were cooled for 24 h at 0~4 °C. After 24 h cooling, the four muscles of loin (*longissimus*), ham, shoulder(*trapezius*), and belly (*obliquus externus abdominis*) were cut from the left half of the carcass using specific methodologies: (I) Pork loin was cut parallel to the junction of the fifth and sixth thoracic vertebrae and sacral vertebrae, 4~6 cm away from the backbone, and the fat layer was slightly trimmed. (II) Pork ham was cut perpendicularly from the *longissimus dorsi* at the junction of the thoracic and sacral vertebrae, leaving it intact. (III) Pork shoulder was cut from the middle of the fifth and sixth ribs. (IV)Pork belly was cut from the fifth and sixth thoracic vertebrae to the lumbar sacral vertebrae junction of the pig, and the rib meat was removed. Then, four muscles were frozen at −20 °C for 48 h.

A total of 37 kinds of mixed fatty acid methyl ester standard and disodium inosinate standard were purchased from Sigma Inc. (St. Louis, MO, USA); the mixed amino acid standard solution was purchased from the Research Center for Agricultural Product Quality Standards of the Ministry of Agriculture and Rural Affairs (Beijing, China), and the cholesterol standard was purchased from the Chinese Academy of Metrology (Beijing, China).

### 2.2. Experimental Methods

#### 2.2.1. Pork Cooking

Pork samples were thawed at 4 °C. The first cooking method was to cut the pork into 0.5 cm-thick pieces and boil them in 100 °C boiling water for 3 min; the second cooking method was to cut the pork into 0.1 cm-thick pieces and scald them in 100 °C boiling water for 10 s; the third method was to cut the pork into 0.5 cm-thick pieces, roast them at 200 °C for 10 min, and turn them over at 5 min. No seasoning was added in the above cooking methods.

#### 2.2.2. Sensory Evaluation Methods

The descriptive sensory evaluation method adopted the sensory profile examination and was performed by 10 professionals who had been especially trained and examined. According to GB/T 22210-2008 “Standard for the Sensory Assessment of Meat and meat Products” [14], ten professionals who have been specially trained and assessed were selected for sensory evaluation. The color, odor, flavor, tenderness, juiciness, chewability, and overall acceptability of the cooked pork were evaluated. Flesh color and marbling were evaluated by referring to the pig color grading atlas [15] and pork marbling grading atlas [16], and the rest of the sensory scores were given 1~10 points from the worst to the best.

#### 2.2.3. Edible Quality Index Determination

Edible quality indexes of fresh pork were determined by referring to NY/T 2793-2015 “Objective Evaluation Method of Meat Quality” [17]. The shearing force was determined by a digital muscle tenderness meter, the meat color values *L**, *a**, and *b** were determined by a portable color difference meter, and water retention was expressed by drip loss.

#### 2.2.4. Nutritional Quality Index Determination

The protein and fat content of fresh pork were determined according to the national standards GB 5009.5-2016 “Determination of Protein in Food under National Food Safety Standard” [18] and GB 5009.6-2016 “Determination of Fat in Foods under National Food Safety Standard” [19].

The amino acid content of fresh pork was determined by referring to GB 5009.124-2016 “Determination of Amino Acids in Food under National Food Safety Standard “ [20], and calculated by the external standard method through the peak area.

The fatty acid content of fresh pork was determined according to GB 5009.168-2016 “Determination of Fatty Acids in Food under National Food Safety Standard “ [21], which was extracted by acetyl chlorine-methanol, with 37 kinds of mixed fatty acid methyl esters used for external standard quantitative analysis.

The inosinic acid content of fresh pork was determined by high-performance liquid chromatography (HPLC) and quantified by the external standard method proposed by Ge et al. [22].

The cholesterol content of fresh pork was determined according to the national standard GB 5009.128-2016 “Determination of Cholesterol in Food under National Food Safety Standard” [23], with slight modification. HPLC was used for determination and the external standard method was used for quantification.

### 2.3. Statistical Analysis

There were two parallels for sensory evaluation, determination of edible quality indexes, protein and inosinic acid, and three parallel for other indexes. Data were processed by EXCEL 2016 (Microsoft, Redmond, WA, USA) to calculate the mean value and standard deviation; SPSS Statistics 26 software (IBM, Chicago, IL, USA) was employed for standardization processing, one-way ANOVA, cluster analysis, correlation analysis, and PCA; Duncan’s multiple comparison was used for difference significance tests, and the significance level was *α =* 0.05. The correlation analysis, cluster analysis, PCA, and coefficient of variation were considered comprehensively, and key evaluation indexes were selected. The range transformation method was applied to the selected indexes for positive processing.

## 3. Results

### 3.1. Fresh Meat Quality Analysis for Different Muscles of Pork

Flesh color, elasticity, and marbling of fresh meat are usually used by consumers to make pork purchase decisions [24], which play an important role in meat quality. Water retention has a significant impact on edible quality, nutrient loss, and muscle production [25], and the shearing force reflects the tenderness of meat [26]. Fat, protein, cholesterol, and fatty acid are nutritional indicators of pork, reflecting the nutritional characteristics of pork [27]. Inosine acid and flavor amino acid promote the formation of pork flavor and are important flavor substances of pork [28]. Flavor amino acids generally refer to glutamic acid, aspartic acid, phenylalanine, alanine, glycine, and tyrosine; these six kinds of amino acids can show special flavor.

As can be seen from Table 1, the flesh color score and marbling score of pork shoulder were significantly higher than those of other muscles (*p <* 0.05). The high flesh color score of the shoulder is due to the high myoglobin content [29]. Marbling can characterize the intramuscular fat content, and an increase in intramuscular fat has a positive significance for pork sensory quality [30].

The *L**, *a**, and *b** values of meat were different among the four muscles of pork. The *L** value of pork loin was significantly higher than that of the other muscles (*p <* 0.05); the *a** values of pork belly and shoulder were significantly higher than that of pork loin and ham (*p <* 0.05), and the *b** value of belly was significantly higher than that of loin and ham (*p <* 0.05). The *L** value difference between the loin and other muscles may be caused by different light scattering characteristics resulting from different fiber structures of the muscle surface [10]; the significantly high *a** value of the belly and shoulder may be due to the rich myoglobin content [29], and the difference in the *b** value may be attributed by the difference in the content of high ferric myoglobin [10,31].

Drip loss of the pork loin was significantly higher than that of other muscles (*p <* 0.05), and the belly had the best water retention, which is consistent with the research results of Damian Knecht et al. [32]. The fat content of the belly was significantly higher than that of other muscles (*p <* 0.05). Cholesterol in the shoulder was higher than that in other muscles (*p <* 0.05), and the inosinic acid, flavored amino acid, and protein content in pork loin were higher than those in other muscles (*p <* 0.05). The content of unsaturated fatty acids (UFA) in pork ham and belly was higher than that in other muscles (*p <* 0.05). There was no significant difference in the content of UFA between pork loin and shoulder, which is consistent with the research results of Huang et al. [33].

### 3.2. Sensory Analysis for the Cooking Quality of Different Muscles of Pork

The four muscles of pork were cooked by boiling, scalding, and roasting, and the sensory quality differences of the four muscles were analyzed by sensory scoring of the cooked meat products.

As can be seen from Figure 1a, the scores of odor, flavor, tenderness, juiciness, and chewiness of belly were significantly higher than those of other muscles under the boiling method (*p <* 0.05). The high odor and flavor scores of the belly may be due to the high fat content of the belly, which produces rich flavor substances through lipid oxidative degradation [28,34,35]. The high juiciness score of the belly is attributed to its good water retention and rich fat content [36].

It can be seen from Figure 1b that under the scalding method the scores of the odor, tenderness, juiciness, and masticatory indexes of pork belly were also significantly higher than those of other muscles (*p <* 0.05). Similar to the boiling method, the key factor that makes the sensory score of pork belly superior to other muscles may be its rich fat content.

As shown in Figure 1c, the scores of cooked flesh color, flavor, tenderness, chewiness, and overall acceptability of shoulder and belly were significantly higher than those of other muscles under the roasting method (*p <* 0.05), and the juiciness scores of belly were significantly higher than those of other muscles (*p <* 0.05), which was also related to the rich fat content of the belly [36]. The odor score of the belly was significantly higher than that of loin and ham (*p <* 0.05), because the Maillard reaction would also occur in the process of roasting besides lipid oxidation [37,38,39], and the greater the degree of Maillard reaction the stronger the aroma [40].

### 3.3. Key Quality Index Screening and Evaluation Model Construction

#### 3.3.1. Principal Component Analysis

To screen the key quality indexes of pork under the methods of boiling, scalding, and roasting, PCA was conducted for each quality index. The results of PCA and the weight of each index under the three cooking methods are shown in Table 2, Table 3 and Table 4.

As can be seen from Table 2, under the boiling method, the quality indexes of different pork muscles were separately analyzed for the four principal components, each of which had a characteristic root greater than 1, and the cumulative variance contribution rate reached 81%. The three-dimensional(3D) loading diagram under boiling method can be seen in Appendix A. The absolute weights of flesh color, elasticity, marbling, *L**, *b**, drip loss, shearing force, cholesterol, inosinic acid, and UFA were lower than others, so they were not considered in the selection of indicators.

As indicated in Table 3, under the scalding method, the quality indexes of different pork muscles were analyzed for the four principal components, each of which had a characteristic root greater than 1, and the cumulative variance contribution rate reached 81%. The 3D loading diagram under scalding method can be seen from Appendix A. The absolute weights of flesh color, elasticity, marbling, *L**, drip loss, shearing force, cholesterol, inosinic acid, and UFA were lower than others, so they were not considered in the selection of indicators.

As can be seen from Table 4, the quality indexes of different pork muscles were analyzed for five principal components under the roasting method, each of which had a characteristic root greater than 1, and the cumulative variance contribution rate reached 82%. The 3D loading diagram under roasting method can be seen from Appendix A. The absolute weights of *L**, *a**, *b**, fat, cholesterol, and UFA were lower than others, so they were not considered when indexes were selected.

#### 3.3.2. Cluster Analysis

To determine key quality indicators of pork under different cooking methods, cluster analysis was performed for all the quality indicators and the results were shown in Figure 2.

Under the boiling method, the dendrogram of the quality indexes was shown in Figure 2a. According to the clustering distance, tenderness, chewiness, juiciness, odor, flavor, fat content, *a**, *b**, and UFA were clustered into a category, while marbling, cholesterol, flesh color, and elasticity were clustered into another. The protein content, flavored amino acid, drip loss, inosinic acid, and *L** were classified into a category, while the shearing force was classified into another category alone.

Under the scalding method, the dendrogram of the quality indexes was shown in Figure 2b. According to the clustering distance, tenderness, chewiness, juiciness, flavor, fat content, odor, *a**, and *b** were grouped into a category, UFA was grouped into a category alone, and marbling, cholesterol, flesh color, and elasticity were grouped into a category. The protein content, flavored amino acid, drip loss, inosinic acid, and *L** were classified into a category, while the shearing force was classified into a category alone.

Under the roasting method, the dendrogram of the quality indexes was shown in Figure 2c. According to the clustering distance, tenderness, juiciness, chewiness, cooked flesh color, fat content, odor, flavor, *a**, and *b** were grouped into a category, UFA was grouped into a category alone, and marbling, cholesterol, flesh color, and elasticity were grouped into a category. Protein, flavored amino acid, drip loss, inosinic acid, and *L** were classified into a category, while the shearing force was classified into another category alone.

#### 3.3.3. Correlation Analysis

To determine key quality indicators of pork under different cooking methods, correlation analysis was carried out for each quality index and the results were shown in Figure 3.

As shown in Figure 3a, under the boiling method, tenderness was significantly positively correlated with chewiness and juiciness, with correlation coefficients of 0.967 and 0.868, respectively (*p <* 0.01). Tenderness is mainly related to myofibrillar protein and connective tissue protein [41]; Sasaki et al. [42] found that tenderness was related to chewiness, juiciness and hardness through experiments. The fat was significantly negatively correlated with the content of flavored amino acid and protein, and the correlation coefficients were −0.920 and −0.904, respectively (*p <* 0.01). Amino acid are the building blocks of proteins. Proteins are broken down to produce amino acids. The content of flavored amino acid was significantly positively correlated with that of protein, and the correlation coefficient was 0.925 (*p <* 0.01).

As shown in Figure 3b, under the scalding method, tenderness was significantly positively correlated with chewiness and juiciness, with correlation coefficients of 0.957 and 0.923, respectively (*p <* 0.01). The content of fat was significantly negatively correlated with the content of flavored amino acid and protein, with the correlation coefficients of −0.920 and −0.904, respectively (*p <* 0.01). The content of flavored amino acid was significantly positively correlated with that of protein, with the correlation coefficient of 0.925 (*p <* 0.01).

It can be seen from Figure 3c that, under the roasting method, flavor and odor were significantly positively correlated (r = 0.839, *p <* 0.01). Tenderness was significantly positively correlated with chewiness (r = 0.921, *p <* 0.01) and juiciness (r = 0.942, *p <* 0.01). With the loss of water, myoglobin and soluble pigment in the muscle are lost [43], so drip loss was positively correlated flesh color (r = −0.769, *p <* 0.01). Drip loss was also positively correlated with inosinic acid (r = 0.899, *p <* 0.01) and protein (r = 0.832, *p <* 0.01).

#### 3.3.4. Screening of Key Quality Indicators

Key quality indicators were screened according to the weight, correlation coefficient, and coefficient of variation of each quality indicator obtained by PCA.

When key quality indexes were screened under the boiling method, the first type of index included tenderness, chewiness, juiciness, odor, flavor, fat, *a**, *b**, and UFA. As *b** and UFA had a small weight, they were not considered. Tenderness was positively correlated with chewiness and juiciness, with the correlation coefficients of 0.967 and 0.868 (*p <* 0.01), coefficient of variation of 18.14%, and weight of 0.2250. Therefore, tenderness was selected as a key index. The weights of odor, flavor, and fat were 0.2162, 0.2298, and 0.1820, respectively, with the coefficients of variation of 11.44%, 14.14%, and 120.57%. Due to the large weights and coefficients of variation of odor, flavor, and fat, the three indexes were also selected. The second type of indicator was marbling, cholesterol, flesh color, and elasticity. Because the weights of the four indicators were relatively low, the second type of indicator was not selected. The third type was the protein content, flavored amino acid, drip loss, inosinic acid, and *L**. Due to the low weights of drip loss, inosinic acid, and *L**, these indicators were not selected. The weights of protein and flavored amino acid were −0.1517 and −0.1600, their coefficients of variation were 20.26% and 25.71%, and the correlation between them was 0.925 (*p <* 0.01). Therefore, flavored amino acid was selected as the third representative index. The fourth type of index was the shearing force, whose weight was too low (−0.0996) so it was not considered. Since the fat was significantly correlated with flavored amino acid, with a correlation coefficient of −0.920, the fat with a higher weight and coefficient of variation was selected to replace flavored amino acid. To sum up, *a**, fat, odor, tenderness, and flavor were selected as the key quality indexes under the boiling method, and their weights were reshown in Table 5.

In the screening of key quality indexes for scalded pork in a hot pot, the first type of index included tenderness, chewiness, juiciness, flavor, fat content, odor, *a**, and *b**. Because tenderness was positively correlated with chewiness and juiciness, with correlation coefficients of 0.957 and 0.923 (*p <* 0.01), weight of 0.2086, and coefficient of variation of 15.29%, tenderness was selected as a key index. The correlation between *b** and *a** was 0.777 (*p <* 0.01), the weight of *a** was 0.1479, and its coefficient of variation was 27.36%. Therefore, *a** was selected as a key index. The correlation between odor and flavor was 0.840 (*p <* 0.01), and the weights of odor and flavor were 0.2073 and 0.2243, both of which were large. Therefore, odor and flavor were selected as key indicators. In short, tenderness, odor, flavor, fat, and *a** were selected to represent the first category. The second type of index was UFA, and because of its low weight of 0.0692 this type of index was deleted. The third category of indicators was marbling, cholesterol, flesh color, and elasticity, and because these four indicators had low weights this category of indicators was not chosen either. The fourth type of index was the protein content, flavored amino acid, drip loss, inosinic acid, and *L**. As the weights of *L**, drip loss, and inosinic acid were relatively low, 0.0516, −0.1147, and −0.1065, respectively, these three indexes were deleted. In this category, the weights of protein and flavored amino acid were −0.1452 and −0.1481, their coefficients of variation were 20.26% and 25.71%, and the correlation between them was 0.925 (*p <* 0.01). Therefore, flavored amino acid was selected as the fourth representative index. The fifth index was the shearing force, whose weight was too low at −0.0691, so it was not considered. Since the fat was significantly correlated with flavored amino acid, with a correlation coefficient of −0.920, the fat with a larger weight and coefficient of variation was selected to replace flavored amino acid. In summary, *a**, fat, odor, tenderness, and flavor were selected as the key evaluation indexes under the scalding method, as shown in Table 5.

When key quality indexes under the roasting method were screened, the first type of index was tenderness, juiciness, chewiness, cooked flesh color, fat content, odor, flavor, *a**, and *b**. Due to the low weights of *a**, *b**, and fat, these indexes were deleted. Tenderness was positively correlated with chewiness (r = 0.921, *p <* 0.01) and juiciness (r = 0.942, *p <* 0.01), its weight was 0.1620, and its coefficient of variation was 13.94%. Therefore, tenderness was selected as a key index. There was a significant positive correlation between flavor and odor (r = 0.839, *p <* 0.01), the weight of flavor was 0.1379, and its coefficient of variation was 11.23%. Therefore, flavor was selected as a key index. In short, tenderness, cooked flesh color, and flavor were selected to represent the first category of indicators. The second type of index was UFA, and this type of index was deleted due to its low weight of 0.0692. The third type was marbling, cholesterol, flesh color, and elasticity. The weight of cholesterol was low, at 0.0963, so this index was excluded. The weight of elasticity was 0.1408, and its coefficient of variation was 11.54%. The weight of marbling was 0.1395, and its coefficient of variation was 14.91%. The weight of flesh color was 0.1653, and its coefficient of variation was 15.3%. Therefore, elasticity, marbling, and flesh color were selected to represent the third type. The fourth type of index was the protein content, flavored amino acid, drip loss, inosinic acid, and *L**. Since *L** had a low weight of −0.0798, this index was not selected. Drip loss was positively correlated with inosinic acid (r = 0.899, *p <* 0.01) and the protein content (r = 0.832, *p <* 0.01). Therefore, drip loss with a larger coefficient of variation was selected to represent the fourth type of index. The fifth type was the shearing force, whose weight, of 0.0111, was too low, so it was not considered. There was a significant negative correlation between cooked flesh color and drip loss (r = −0.769, *p <* 0.01). In order to avoid a negative comprehensive score, cooked flesh color was chosen to represent drip loss. Therefore, six indexes including flavor, marbling, elasticity, cooked flesh color, tenderness, and flesh color were selected as the key indexes under the roasting method, and the weights were reshown in Table 5.

#### 3.3.5. Model Construction and Suitability Evaluation

According to the selected key quality indicators and weights, an evaluation model for the cooking suitability of different muscles of meat was constructed. According to the comprehensive quality evaluation scores and K-means clustering analysis, the suitability of the four muscles of pork under the three cooking methods was obtained and the results were shown in Table 6. The score of comprehensive quality evaluation was set as *Y*. Under the boiling method, *Y* greater than 0.7131 was the most suitable for boiling muscles, 0.3058~0.7131 was relatively suitable for boiling muscles, and less than 0.3058 was unsuitable for boiling muscles. Under the scalding method, *Y* greater than 0.647 was the most suitable, 0.251~0.647 was relatively suitable, and less than 0.251 was unsuitable. Under the roasting method, *Y* greater than 0.4469 was the most suitable for roasting, 0.2344~0.4469 was relatively suitable for roasting, and *Y* less than 0.2344 was not suitable for roasting.

The evaluation model under the boiling method was:(1)$$Y=0.1537X_{1}+0.1805X_{2}+0.2145X_{3}+0.2233X_{4}+0.2281X_{5}$$

*X*_1_~*X*_5_ represent *a**, fat, odor, tenderness, and flavor, respectively. After the determination results of the key quality indicators for boiling were normalization and standardized, they were substituted into the comprehensive quality model to calculate the scores, as shown in Table 6. The results showed that one of the most suitable muscles for boiling was pork belly.

The evaluation model under the scaling method was:(2)$$Y=0.1541X_{1}+0.1787X_{2}+0.2160X_{3}+0.2174X_{4}+0.2337X_{5}$$

*X*_1_~*X*_5_ represent *a**, fat, odor, tenderness, and flavor, respectively. The measured results of the key quality indicators for scalding were normalization and standardized into the comprehensive quality model to calculate the scores, as shown in Table 6. The most suitable muscle for scalding was pork belly.

The evaluation model under the roasting method was:(3)$$Y=0.1539X_{1}+0.1557X_{2}+0.1572X_{3}+0.1677X_{4}+0.1808X_{5}+0.1845X_{6}$$

*X*_1_~*X*_6_ represent flavor, marbling, elasticity, cooked flesh color, tenderness, and flesh color, respectively. The measured results of the key quality indicators of roasting were normalized and standardized, and then substituted into the comprehensive quality model to calculate the scores, as shown in Table 6. The most suitable muscles for roasting were pork belly and shoulder.

### 3.4. Verification of the Comprehensive Quality Evaluation Equation for Different Muscles of Pork

The overall acceptability score in the sensory evaluation experiment was a dependent variable, the score calculated by the comprehensive quality evaluation model was the independent variable, and then a regression equation was established. The validation model under the boiling method was:
*y* = 1.8236*X* − 0.3596 (*R*^2^ = 0.8862)
(4)


The validation model under the scalding method was:
*y* = 1.8388*X* − 0.3479 (*R*^2^ = 0.9591)
(5)


The validation model under the roasting method was:
*y* = 3.5021*X* − 0.6924 (*R*^2^ = 0.8213)
(6)

where *R*^2^ is greater than 0.8, indicating that the comprehensive evaluation model can accurately predict the overall acceptability of sensory evaluation and can accurately reflect the suitability of the three cooking methods.

## 4. Conclusions

This study analyzed the differences in the sensory quality of four muscles of pork in three cooking methods (boiling, scalding, and roasting) and compared the differences in the edible quality and nutritional quality of fresh meat. Through principal component analysis, correlation analysis, and analysis of the coefficient of variation, key quality indexes under different cooking methods were determined and the comprehensive quality evaluation model was established for pork under different cooking methods. Under the boiling method, the comprehensive quality evaluation model was $Y=0.1537X_{1}+0.1805X_{2}+0.2145X_{3}+0.2233X_{4}+0.2281X_{5}$ (*X*_1_~*X*_5_ are *a**, fat, odor, tenderness, and flavor, respectively), and the most suitable muscle was belly. Under the scalding method, the comprehensive quality evaluation model was $Y=0.1541X_{1}+0.1787X_{2}+0.2160X_{3}+0.2174X_{4}+0.2337X_{5}$ (*X*_1_~*X*_5_ are *a**, fat, odor, tenderness, and flavor, respectively), and the most suitable muscle was belly. Under the roasting method, the comprehensive quality evaluation model was $Y=0.1539X_{1}+0.1557X_{2}+0.1572X_{3}+0.1677X_{4}+0.1808X_{5}+0.1845X_{6}$ (*X*_1_~*X*_6_ are flavor, marbling, elasticity, cooked flesh color, tenderness, and flesh color, respectively), and the most suitable muscles were belly and shoulder.

## Figures and Tables

**Figure 1 foods-12-00742-f001:**
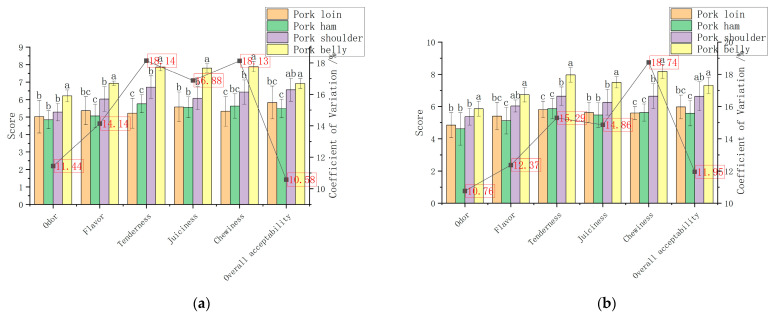
Sensory scores and variation coefficients of each index for four muscles of pork under different cooking methods: (**a**) boiling; (**b**) scalding; (**c**) roasting. Different letters (a, b, c) indicated significant difference (α = 0.05 level).

**Figure 2 foods-12-00742-f002:**
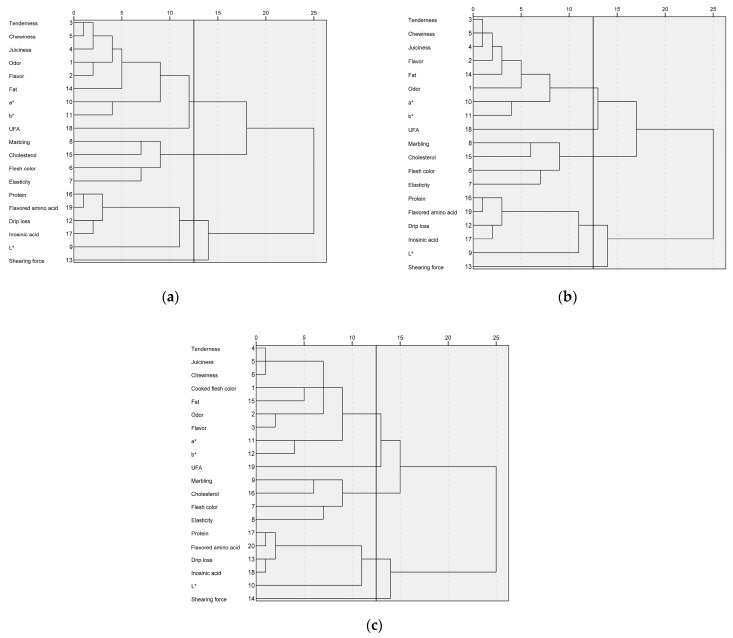
Cluster analysis of the quality indexes of pork under different cooking methods: (**a**) boiling; (**b**) scalding; (**c**) roasting.

**Figure 3 foods-12-00742-f003:**
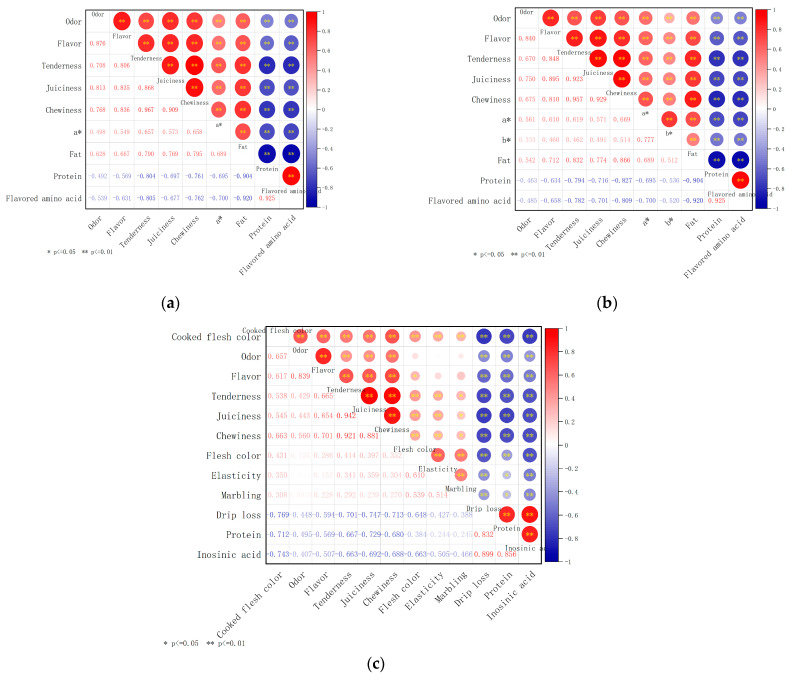
Correlation analysis of certain indexes of pork under different cooking methods: (**a**) boiling; (**b**) scalding; (**c**) roasting. * is significantly correlated (*α =* 0.05 level), ** is extremely significantly correlated (*α =* 0.01 level).

**Table 1 foods-12-00742-t001:** Analysis of the fresh pork quality results in different muscles.

Indicators	Pork Loin	Pork Ham	Pork Shoulder	Pork Belly	Coefficient of Variation (%)
Flesh color	3.31 ± 0.53 ^c^	3.88 ± 0.44 ^b^	4.79 ± 0.48 ^a^	4.01 ± 0.10 ^b^	15.30
Elasticity	3.94 ± 0.40 ^b^	4.57 ± 0.84 ^ab^	5.23 ± 0.44 ^a^	4.62 ± 0.41 ^ab^	11.54
Marbling	2.57 ± 0.27 ^b^	2.83 ± 0.53 ^b^	3.85 ± 0.42 ^a^	2.72 ± 0.28 ^b^	14.91
*L**	52.25 ± 4.97 ^a^	46.62 ± 3.06 ^b^	46.12 ± 5.32 ^b^	45.65 ± 2.71 ^b^	6.48
*a**	11.41 ± 1.25 ^b^	13.93 ± 3.07 ^b^	19.69 ± 2.82 ^a^	20.75 ± 3.64 ^a^	27.36
*b**	3.16 ± 1.65 ^c^	4.14 ± 1.25 ^bc^	6.00 ± 1.49 ^ab^	6.53 ± 1.79 ^a^	31.78
Drip loss (%)	9.87 ± 1.98 ^a^	7.86 ± 1.85 ^b^	1.41 ± 1.07 ^c^	0.52 ± 0.24 ^c^	94.63
Shearing force (kgf)	5.25 ± 2.29 ^a^	5.70 ± 1.79 ^a^	4.11 ± 0.93 ^a^	4.26 ± 1.05 ^a^	15.92
Protein (%)	21.17 ± 0.43 ^a^	19.13 ± 1.16 ^b^	16.23 ± 1.24 ^c^	13.07 ± 1.66 ^d^	20.26
Fat (%)	1.44 ± 0.54 ^c^	2.00 ± 0.73 ^c^	13.57 ± 2.82 ^b^	34.90 ± 9.96 ^a^	120.57
Cholesterol (mg/100 g)	50.21 ± 1.61 ^c^	57.53 ± 2.55 ^b^	60.92 ± 2.36 ^a^	49.62 ± 2.51 ^c^	10.18
UFA (%)	58.02 ± 1.07 ^b^	60.12 ± 0.91 ^a^	57.19 ± 1.33 ^b^	61.21 ± 0.80 ^a^	3.14
Inosinic acid (mg/100 g)	226.55 ± 21.49 ^a^	171.50 ± 32.36 ^b^	92.66 ± 16.65 ^c^	94.74 ± 25.83 ^c^	44.30
Flavored amino acid (g/100 g)	9.49 ± 0.36 ^a^	8.85 ± 0.17 ^b^	6.7 ± 0.48 ^c^	5.27 ± 1.05 ^d^	25.71

Means in the same row with different letters differ significantly (*α =* 0.05 level).

**Table 2 foods-12-00742-t002:** Principal component analysis results of boiled pork.

Indicators	PC1	PC2	PC3	PC4	Weight
Odor	0.701	0.462	−0.105	0.340	0.2162
Flavor	0.774	0.366	0.046	0.312	0.2298
Tenderness	0.921	0.168	0.096	0.11	0.2250
Juiciness	0.831	0.423	−0.036	0.194	0.2335
Chewiness	0.900	0.298	0.121	0.145	0.2418
Flesh color	0.405	−0.752	−0.207	0.041	−0.0277
Elasticity	0.311	−0.603	−0.121	0.387	0.0034
Marbling	0.274	−0.754	0.230	0.289	−0.0001
*L**	−0.291	0.375	0.639	0.392	0.0700
*a**	0.816	−0.165	0.178	−0.085	0.1549
*b**	0.616	−0.116	0.494	−0.053	0.1495
Drip loss	−0.867	0.318	0.083	0.043	−0.1264
Shearing force	−0.368	0.016	−0.666	0.44	−0.0996
Fat	0.910	0.160	−0.083	−0.244	0.1820
Cholesterol	−0.109	−0.831	0.008	0.151	−0.1165
Protein	−0.914	0.076	0.099	0.241	−0.1517
Inosinic acid	−0.856	0.400	0.115	0.043	−0.1111
UFA	0.356	0.491	−0.546	−0.018	0.0870
Flavored amino acid	−0.925	0.090	0.051	0.187	−0.1600
Characteristic root	9.181	3.551	1.614	1.055	
Variance contribution rate (%)	48.323	18.692	8.496	5.555	
Cumulative variance contribution rate (%)	48.323	67.015	75.511	81.066	

**Table 3 foods-12-00742-t003:** Principal component analysis results of scalded pork.

Indicators	PC1	PC2	PC3	PC4	Weight
Odor	0.669	0.291	0.111	0.548	0.2073
Flavor	0.806	0.324	0.146	0.326	0.2243
Tenderness	0.889	0.304	0.025	0.046	0.2086
Juiciness	0.850	0.349	0.137	0.218	0.2272
Chewiness	0.926	0.247	0.014	0.042	0.2081
Flesh color	0.425	−0.757	−0.180	−0.006	−0.0196
Elasticity	0.338	−0.594	−0.046	0.361	0.0186
Marbling	0.333	−0.711	0.326	0.285	0.0286
*L**	−0.300	0.383	0.666	0.178	0.0516
*a**	0.821	−0.128	0.159	−0.090	0.1479
*b**	0.631	−0.057	0.463	−0.143	0.1402
Drip loss	−0.863	0.302	0.105	0.098	−0.1147
Shearing force	−0.357	−0.018	−0.539	0.623	−0.0691
Fat	0.911	0.211	−0.148	−0.198	0.1715
Cholesterol	−0.084	−0.852	0.076	0.091	−0.1001
Protein	−0.920	0.028	0.159	0.200	−0.1452
Inosinic acid	−0.861	0.375	0.131	0.062	−0.1065
UFA	0.314	0.474	−0.596	0.044	0.0692
Flavored amino acid	−0.923	0.052	0.106	0.188	−0.1481
Characteristic root	9.282	3.293	1.615	1.255	
Variance contribution rate (%)	48.854	17.33	8.499	6.607	
Cumulative variance contribution rate (%)	48.854	66.184	74.683	81.289	

**Table 4 foods-12-00742-t004:** Principal component analysis results of roasted pork.

Indicators	PC1	PC2	PC3	PC4	PC5	Weight
Cooked flesh color	0.8380	−0.0550	0.0610	−0.0920	0.2750	0.1502
Odor	0.6060	−0.3930	0.2260	0.2570	0.5250	0.0981
Flavor	0.7260	−0.2270	0.1350	0.4210	0.3160	0.1379
Tenderness	0.8040	−0.0410	0.0530	0.4120	−0.3440	0.1620
Juiciness	0.8320	−0.1080	0.1150	0.3020	−0.3630	0.1578
Chewiness	0.8470	−0.1420	−0.0070	0.3580	−0.1860	0.1523
Flesh color	0.5590	0.6350	0.2160	−0.0440	−0.0540	0.1653
Elasticity	0.4240	0.5580	0.1550	0.1080	−0.1340	0.1408
Marbling	0.3980	0.7330	−0.1150	0.1960	0.2240	0.1395
*L**	−0.3490	−0.2550	−0.5000	0.5850	0.0550	−0.0798
*a**	0.7840	0.0340	−0.2870	−0.2810	0.3010	0.0977
*b**	0.5960	0.0020	−0.5630	−0.0540	0.2440	0.0583
Drip loss	−0.9170	−0.1450	−0.0280	0.0740	0.0710	−0.1655
Shearing force	−0.3140	0.0210	0.7620	0.1450	0.2210	0.0111
Fat	0.8570	−0.3590	−0.0600	−0.2460	−0.0510	0.0917
Cholesterol	0.0380	0.8640	0.1000	0.0480	0.1590	0.0963
Protein	−0.9050	0.1260	0.0210	0.2330	0.0890	−0.1249
Inosinic acid	−0.9080	−0.2250	−0.0440	0.1500	0.0570	−0.1676
UFA	0.2720	−0.5910	0.4500	−0.2320	−0.0040	0.0077
Flavored amino acid	−0.9160	0.0990	0.0610	0.1960	0.1100	−0.1288
Characteristic root	9.627	2.882	1.626	1.377	1.065	
Variance contribution rate (%)	48.135	14.412	8.129	6.883	5.325	
Cumulative variance contribution rate (%)	48.135	62.547	70.676	77.559	82.884	

**Table 5 foods-12-00742-t005:** Weight of key quality indicators.

Method of Cooking	Key Indicator	PCA Weight	Normalized Weight
Boiling	*a** (*X*_1_)	0.1549	0.1537
	Fat (*X*_2_)	0.1820	0.1805
	Odor (*X*_3_)	0.2162	0.2145
	Tenderness (*X*_4_)	0.2250	0.2233
	Flavor (*X*_5_)	0.2298	0.2281
Scalding	*a** (*X*_1_)	0.1479	0.1541
	Fat (*X*_2_)	0.1715	0.1787
	Odor (*X*_3_)	0.2073	0.2160
	Tenderness (*X*_4_)	0.2086	0.2174
	Flavor (*X*_5_)	0.2243	0.2337
Roasting	Flavor (*X*_1_)	0.1379	0.1539
	Marbling (*X*_2_)	0.1395	0.1557
	Elasticity (*X*_3_)	0.1408	0.1572
	Cooked flesh color (*X*_4_)	0.1502	0.1677
	Tenderness (*X*_5_)	0.1620	0.1808
	Flesh color (*X*_6_)	0.1653	0.1845

**Table 6 foods-12-00742-t006:** Comprehensive quality evaluation scores of four pork muscles under three cooking methods and K-means cluster analysis results.

Method of Cooking	Muscle	*Y*	Suitability
Boiling	Pork loin	0.2774	Unsuitable
	Pork ham	0.2967	Unsuitable
	Pork shoulder	0.5212	Relatively suitable
	Pork belly	0.7841	The most suitable
Scalding	Pork loin	0.2602	Relatively suitable
	Pork ham	0.2505	Unsuitable
	Pork shoulder	0.4965	Relatively suitable
	Pork belly	0.7449	The most suitable
Roasting	Pork loin	0.1967	Unsuitable
	Pork ham	0.2573	Relatively suitable
	Pork shoulder	0.4484	The most suitable
	Pork belly	0.4249	The most suitable

## Data Availability

The data presented in this study are available on request from the corresponding author. The data are not publicly available due to involving other research content.

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
