# Peer review of "The Establishment of Evaluation Models for the Cooking Suitability of Different Pork Muscles"

_foods, 2023, doi:10.3390/foods12040742_

Round 1

Reviewer 1 Report

Introduction:

Please discuss the pork with reference to the global production and needs. Foods Journal is not specific to China. Re-write introduction by including pork production across the world with a specific focus on China’s pork production.

Why Pork is preferred by Chinese people??? Please give reason here? Nutritional quality vs. cost effective

Line 35: Please replace “Boiling pork” with “boiled pork”

Line 25: Please replace the keywords with some suitable words. As they already appeared in the manuscript title so, no need to repeat these words here.

pork; muscles; cooking suitability; evaluation model

Line 36: Tenderloin is the name of the cut not name of the muscle. Please write the muscle name here.

Line 37: Write muscle names of shoulder and belly and include throughout the paper.

Line 72: Write the name of muscle present in Loin region.

Line 69: replace “6” with “six”

I think, six animals are not enough for this type of study

What was the gender of these six pigs slaughtered?

Line 71: Did these animals were stunned before slaughtering?

What was the pH at the time of deboning/cutting of carcasses?

Please write specifications of the chiller

Line 73: Why there is a need to freeze the samples? Freezing can negatively impact meat quality. Please write specifications of the freezer

Line 81: How much steaks were used per cooking treatment??

Line 88: Please explain sensory analysis. Number of panelists? Their age? Number of cooking sessions and testing sessions?

If the panelists are well trained, why there is a need to calculate overall acceptability? Overall acceptability is not needed. Please remove it.

Conclusion: Need to explain it more. 

Reviewer 2 Report

Although it is an important study especially in terms of sensory properties, it is important to answer the following questions in order to elaborate the material method part a little more.

How was the number of samples specified in the supplementary section determined?

How many trained panelists participated? In qualitative studies, it is especially useful to specify the number of panelists. 

How many samples were analyzed?

As is well known, the size of the data sets in such prediction studies is important for the validity of the study. 

Reviewer 3 Report

Dear Authors,

The study has a relevant scope by constructing mathematical models for the evaluation of quality of pork meat. Some comments and suggestions are indicated below.

Lines 28-31: References for each statement are missing

Lines 35-38: References for each statement are missing

Section 2.3: The number of replications of the entire experiment is missing. Cluster analysis is missing.

Lines 129-135: Each one of these statements must be supported by a reference

Line 154. What means “flavored amino acid”?

Line 175: Please delete “cooked”

Section 3.3.1. Principal component analysis. Eigenvalues results is missing. It is not possible to interpret the information presented in these tables without the plots. This section does not have discussion. Please revise it.

Section 3.3.2. Cluster analysis. The relevance of using cluster analysis in dependent variables instead of independent variables is not clear. Cluster analysis is a key tool to group samples in terms of their properties, but not the other way around. Discussion of data is missing.

Section 3.3.3. Correlation analysis. Discussion is missing. Essentially, why one variable correlated with another variables? What physical, chemical, or biological mechanism explains this specific correlation? This line of thought has to be applied to each significant correlation within each cooking method.

3.3.5. Model construction and suitability evaluation. Have the coefficients for each model been tested for significance?

Table 1. Heading could be “Analysis of the fresh pork quality results in different muscles”.

Table 1. Coefficient of variation for each variable does not reveal any relevant information. Values above 100% does not make sense. Either remove this column or calculate the standard error of mean.

Tables 2, 3 and 4. Graphical representation of PCA data is missing.

Figure 1. Please remove the Coefficient of Variation (values indicated with red color)

Round 2

Reviewer 1 Report

You didn't mention the line numbers where you made necessary changes.

Line 75: The slaughtered pigs were placed in a cold storage at 0~4 ° C 75 for 24h to cool and remove acid

What do you mean by remove acid? It's incorrect. Do you mean to say acidification? Please clarify here.

Line 76-78: Please write complete and correct names of muscles. Like, longissimus muscle starts from last cervical vertebrae and ends near last lumber vertebrae. Please specify specific part of muscles here.

Two steaks were used for per cooking treatment.

I don't think its enough to reach at a conclusion. How you can average it for statistical analysis? 

The sensory analysis team consisted of 10 members, aged 20-50 years old. Why there is huge difference in the age of the sensory panelists?  It's not a large-scale consumer-based study where hundreds of people were used to analyze it. For only 10 panelists, age difference is huge, please verify it.  

Why there is a need to get overall acceptability? When you are considering all other sensory characteristics, it is not needed.  

Reviewer 2 Report

I thank the authors for their contributions to the literature. 

Author Response

感谢您的评论。

Reviewer 3 Report

Dear Authors,

The manuscript was properly improved.

Author Response

感谢您的评论。